# Advancing Glaucoma Care: Integrating Artificial Intelligence in Diagnosis, Management, and Progression Detection

**DOI:** 10.3390/bioengineering11020122

**Published:** 2024-01-26

**Authors:** Yan Zhu, Rebecca Salowe, Caven Chow, Shuo Li, Osbert Bastani, Joan M. O’Brien

**Affiliations:** 1Department of Ophthalmology, Scheie Eye Institute, University of Pennsylvania, Philadelphia, PA 19104, USA; yan.zhu1@pennmedicine.upenn.edu (Y.Z.); rebecca.salowe@pennmedicine.upenn.edu (R.S.); caven.chow@pennmedicine.upenn.edu (C.C.); 2Department of Computer & Information Science, University of Pennsylvania, Philadelphia, PA 19104, USA; lishuo1@seas.upenn.edu (S.L.); obastani@seas.upenn.edu (O.B.)

**Keywords:** artificial intelligence, glaucoma, computer-aided diagnosis, screening, precision medicine, machine learning

## Abstract

Glaucoma, the leading cause of irreversible blindness worldwide, comprises a group of progressive optic neuropathies requiring early detection and lifelong treatment to preserve vision. Artificial intelligence (AI) technologies are now demonstrating transformative potential across the spectrum of clinical glaucoma care. This review summarizes current capabilities, future outlooks, and practical translation considerations. For enhanced screening, algorithms analyzing retinal photographs and machine learning models synthesizing risk factors can identify high-risk patients needing diagnostic workup and close follow-up. To augment definitive diagnosis, deep learning techniques detect characteristic glaucomatous patterns by interpreting results from optical coherence tomography, visual field testing, fundus photography, and other ocular imaging. AI-powered platforms also enable continuous monitoring, with algorithms that analyze longitudinal data alerting physicians about rapid disease progression. By integrating predictive analytics with patient-specific parameters, AI can also guide precision medicine for individualized glaucoma treatment selections. Advances in robotic surgery and computer-based guidance demonstrate AI’s potential to improve surgical outcomes and surgical training. Beyond the clinic, AI chatbots and reminder systems could provide patient education and counseling to promote medication adherence. However, thoughtful approaches to clinical integration, usability, diversity, and ethical implications remain critical to successfully implementing these emerging technologies. This review highlights AI’s vast capabilities to transform glaucoma care while summarizing key achievements, future prospects, and practical considerations to progress from bench to bedside.

## 1. Introduction

Glaucoma, often referred to as the “silent thief of sight”, is the leading cause of irreversible blindness worldwide [1]. Its insidious nature, characterized by a gradual loss of peripheral vision often unappreciated by the patient, underscores the critical importance of early detection and continuous monitoring [2]. In the clinic, glaucoma is typically diagnosed and monitored using a multimodal approach, including tonometry to measure intraocular pressure (IOP), visual field tests, optical coherence tomography (OCT), and fundoscopic examinations [3]. These methods, while foundational, have their limitations: tonometry can be influenced by corneal thickness, visual field tests depend on patient responsiveness, and OCT and fundoscopic exams require expert interpretation, often with some degree of subjectivity [4]. Given these constraints, as the emphasis on early detection and intervention grows, there is an unmet need for more consistent, objective, and precise monitoring techniques [5]. Artificial intelligence (AI) has emerged as a solution to harness this extensive data, aiming to offer automated, consistent, and predictive insights into all areas of glaucoma care [6].

AI’s ability to analyze vast amounts of data, detect intricate patterns, and predict disease trajectories offers a paradigm shift in how we approach glaucoma diagnosis and monitoring [7]. This review paper delves into the transformative role of AI in reshaping glaucoma care, highlighting its methodologies, impacts, challenges, and future potential. In tracing the historical context, the journey of AI’s integration into glaucoma care is a testament to the continuous evolution of medical technology. The late 20th and early 21st centuries saw a surge in ophthalmic imaging techniques, notably OCT, providing high-resolution views of the optic nerve and retinal layers [8]. With this influx of data came increased challenges in interpretation. It was during this phase, particularly in the 2010s, that AI began making its mark. Leveraging machine learning algorithms, early applications of AI sought to automate the analysis of visual fields and OCT scans, aiming to identify subtle patterns indicative of glaucoma progression [9]. Transitioning through the years, as datasets grew and algorithms became more sophisticated, AI’s role transitioned from simple analysis to prediction, including forecasting disease trajectories and potential treatment outcomes.

In light of these developments, this review focuses on the multifaceted applications and implications of AI in glaucoma care, which are illustrated in Figure 1. At the core of this schematic is the AI continuum, representing various AI methodologies such as Machine Learning (ML), Neural Networks (NN), and Deep Learning (DL), which form the foundation for advanced data analysis in glaucoma research. We explore the transformative potential of AI in enhancing the impact of screenings, improving diagnostic accuracy, personalizing treatment strategies, and improving patient outcomes. This paper also examines the integration of AI in precision medicine and its potential in patient education, counseling, and medication adherence. Additionally, we discuss the challenges and considerations associated with implementing AI in clinical settings, emphasizing the need for rigorous validation and addressing data integrity concerns. As the landscape of glaucoma care evolves, this review underscores the pivotal role of AI in shaping its future, offering insights into both its promising advancements and the hurdles that lie ahead.

## 2. From Traditional to AI-Enhanced Data Collection

Data collection for glaucoma diagnosis has traditionally revolved around a combination of clinical assessments and specialized imaging techniques. Key metrics such as IOP are gathered using tonometry, while the structure of the optic nerve head and retinal nerve fiber layer (RNFL) are visualized through imaging modalities such as OCT and fundus photography. Additionally, visual field tests map out the patient’s field of vision, identifying any deficits or abnormalities characteristic of glaucoma. Recognizing the limitations of traditional methods, as the volume of diagnostic data grew, so did the need for more efficient and accurate data processing.

The integration of AI is beginning to transform the data collection process in glaucoma care, enhancing both efficiency and accuracy. For example, AI-powered imaging devices are now being developed to auto-calibrate based on patient specifics, which could potentially improve image quality [10]. Additionally, emerging AI algorithms in these devices process data in real-time, providing immediate insights and the ability to predict trends based on historical data [11]. The Retinal Fundus Glaucoma Challenge (REFUGE) represents a pivotal step in AI-driven ophthalmology [12]. Established with MICCAI 2018, it addressed the constraints of conventional glaucoma assessment using color fundus photography. REFUGE introduced a groundbreaking dataset of 1200 fundus images with detailed ground truth segmentations and clinical labels, the largest of its kind. This initiative was crucial for standardizing AI model evaluations in glaucoma diagnosis, allowing for consistent and fair comparisons. Notably, some AI models in the challenge surpassed human experts in glaucoma classification, demonstrating AI’s potential in enhancing diagnostic precision through an advanced, large-scale dataset [12]. Although this marks a significant advancement, the field is still in the early stages of transitioning into the AI era, with traditional methods and AI-based approaches coexisting.

This evolution is further augmented by the advent of wearable technology, which introduces new dimensions in continuous monitoring and real-time data analysis for glaucoma management. With the advent of wearable technology and smart devices, continuous monitoring and real-time data collection have become feasible [13]. One of the most useful wearable technologies for glaucoma detection are contact lenses. The SSCLs introduced by Zhang et al. [14] allow continuous 24 h monitoring of IOP through an embedded wireless sensor built upon commercial soft contact lenses. In vivo testing in a dog model demonstrated the ability to wirelessly track circadian IOP fluctuations with a sensitivity of 662 ppm/mmHg (R^2^ = 0.88) using a portable vector network analyzer coupled to a contact-lens reader coil. Measurements in human subjects exhibited even higher sensitivity of 1121 ppm/mmHg (R^2^ = 0.91) attributed to superior fit enabled by the soft hydrogel lens base. This sensitivity exceeds previous wearable sensors by more than two times. The seamless interface of the SSCLs with the cornea was confirmed through anterior segment OCT imaging in human eyes. The wireless, 24 h IOP data obtained by these soft hydrogel-based sensors can aid in glaucoma detection and management through continuous monitoring of ocular hypertensive events and linking IOP trends to disease progression. Thus, from ensuring quality data collection to real-time processing, AI is beginning to embed itself deeply into the data collection process for glaucoma diagnosis and management, making these more robust and insightful.

However, recent studies in AI-assisted glaucoma diagnosis underscore the importance of training data diversity. For example, the REFUGE challenge, a significant initiative in AI-driven ophthalmology, utilized a dataset of 1200 fundus images, aiming for broad demographic representation. However, as highlighted in the literature, there is a recognized need for more inclusive data encompassing a wider range of ethnicities and age groups. This inclusivity is critical, given the variability in glaucoma presentation across different populations. Studies emphasize the potential risk of biased AI models due to non-diverse training datasets, which may not effectively represent glaucoma manifestations in underrepresented groups. Thus, the current shift towards more ethnically and demographically inclusive datasets is a vital step in developing universally applicable and unbiased AI models for glaucoma detection.

## 3. AI’s Role in Glaucoma Screening

### 3.1. Early Detection and Challenges

Early detection and treatment of glaucoma is essential as vision loss from the disease is currently irreversible. However, the disease is difficult to detect in the early stages, as it is asymptomatic and typically begins with peripheral rather than central vision loss. As a result, almost 50% of glaucoma patients are undiagnosed, delaying treatment until irreversible vision loss has already occurred [15]. Screening for glaucoma is therefore an important mechanism to detect signs of disease in undiagnosed individuals, allowing intervention while there is still vision left to preserve [16]. Currently, the impact and reach of glaucoma screenings is limited by a reliance on individual examinations by glaucoma specialists, ophthalmologists, or optometrists. Screenings can be lengthy, labor-intensive, and challenging to practically implement, ultimately limiting the number of screened individuals. This is especially true in developing countries, where there is a high burden of glaucoma and a limited number of trained eye professionals [17]. As the prevalence of glaucoma continues to rise in an aging population, there is a growing mismatch between the need for glaucoma screenings and the supply of available resources [18].

### 3.2. AI’s Potential to Transform Glaucoma Screening

In response to these challenges, AI-enabled screening for glaucoma could help fill this unmet need, increasing access to care and lessening the burden on healthcare systems [19]. A system that accurately flags possible glaucoma on images in real time could allow for large-scale screenings to be conducted without the presence of a vision specialist. Patients with signs of glaucoma could then be referred to an ophthalmologist or optometrist for a comprehensive examination, diagnosis, and treatment. Ideally, glaucoma screenings utilizing AI would be low-cost, accurate, and easily translated to low-resource settings. These screenings could then take place in remote rural areas, underserved urban areas, or countries with a scarcity of ophthalmic specialists providing frontline eye care.

Fundus photography, a low-cost option that fits these criteria, has already been successfully incorporated into AI-enabled screening programs to detect diabetic retinopathy [20]. Fundus images provide visualization of anatomic changes to the optic nerve head, such as optic disc cupping and thinning of the neuroretinal rim. These structural abnormalities often precede loss of visual fields [21]. Among its benefits for screening, fundus photography is low-cost, non-invasive, quick, and portable, allowing application to low-resource settings [22]. Starting around 2018, many studies have developed convolutional neural networks (CNNs) trained on thousands of labeled fundus photos to distinguish glaucomatous from healthy eyes [23]. A range of CNN architectures have been applied including ResNet, InceptionNet, and VGGNet, often utilizing transfer learning and reporting high performance [24].

Recent advancements in glaucoma detection have incorporated Vision Transformers like the data-efficient image transformer (DeiT), showing notable efficacy in analyzing fundus photography. These models utilize self-attention mechanisms, effectively capturing the global characteristics of fundus images and thereby enhancing classification accuracy. For instance, studies such as those by Wassel et al. [25] and Fan et al. [26] have demonstrated the competitive performance of Vision Transformer models, particularly in terms of generalizability across diverse datasets. Notably, the attention maps from DeiT models tend to concentrate on clinically relevant areas, like the neuroretinal rim, aligning with regions commonly assessed in manual image review. This alignment suggests that DeiT models can complement traditional diagnostic approaches by focusing on key areas used in glaucoma assessment. The emerging use of Vision Transformers, including DeiT, in glaucoma detection highlights their potential in contributing to the evolving landscape of AI applications in ophthalmology.

### 3.3. AI Outperforming Human Experts and Challenges

Several studies have shown that deep learning models can achieve equal or better accuracy in differentiating normal from glaucomatous eyes when compared with expert glaucoma specialists [27,28,29,30]. Ting et al. [31] developed a deep learning model using 494,661 retinal images to detect diabetic retinopathy and achieved an AUC of 0.942 in detecting “referable” glaucoma. Similarly, Li et al. [24] created a deep learning algorithm using 48,116 fundus images to detect glaucomatous optic neuropathy. The model achieved an AUC of 0.986, with a sensitivity of 95.6% and specificity of 92.0%. The Pegasus system (version v1.0) [32], a cloud-based AI from Visulytix Ltd. (London, UK) evaluates fundus photos using specialized CNNs to extract and classify the optic nerve. Compared to medical professionals, it achieved an accuracy of 83.4% in identifying glaucomatous damage. Orbis International provides free access to an AI tool called Cybersight AI to eye care professionals in low- and middle-income countries [33]. This open access tool can detect diabetic retinopathy, glaucoma, and macular disease on fundus images. At clinics in Rwanda, screening with this device led to accurate referrals for diabetic retinopathy and high rates of patient satisfaction, though more research is needed on diagnostic accuracy for glaucoma [33].

Several challenges must be addressed in order to successfully integrate AI-enabled glaucoma screening into real-world settings. First, we must ensure that deep learning models maintain their accuracy when applied to images from different cameras with varying photographic quality. Studies show that these models currently underperform when images are captured on different cameras compared with those used in training datasets [34]. Second, more research is needed on how co-morbid pathologies can impact the performance of such algorithms. Anatomic variability and pathologic conditions can affect the appearance of the optic nerve head, so many training datasets eliminate images with ocular pathologies; however, real-world screenings will be filled with individuals with a variety of ocular conditions. Finally, more studies must integrate testing of deep learning models in the actual settings where they will be implemented and ensure generalizability to diverse racial and ethnic groups. Many models with high accuracies upon testing do not demonstrate similar accuracy in the real world [34].

The interpretability of AI models in clinical settings is a crucial aspect that warrants detailed discussion. AI models, particularly those based on deep learning, often function as ‘black boxes’, providing limited insight into how they derive their conclusions. This lack of transparency can be a significant barrier to the adoption of AI in clinical practice, where understanding the reasoning behind a diagnosis is fundamental for clinician trust and decision making.

Recent advancements in AI have seen the development of techniques aimed at unraveling these black boxes, thus enhancing the interpretability of AI systems. Methods such as Layer-wise Relevance Propagation (LRP) and Class Activation Mapping (CAM) are being explored to provide visual explanations of AI decisions. For instance, in glaucoma detection, these methods can highlight areas in fundus images or OCT scans that the AI model deems significant for its diagnosis. This not only aids clinicians in understanding AI decisions but also serves as a tool for validating the accuracy of the AI model. The integration of such interpretability frameworks into AI systems for glaucoma detection is a promising step towards their acceptance and effective utilization in clinical environments.

## 4. AI’s Role in Glaucoma Diagnosis

Unlike screening, where the primary aim of AI is to flag potential glaucoma cases for further examination, AI in glaucoma diagnosis tackles a more nuanced challenge. Here, AI is tasked with confirming the presence of glaucoma in individuals who have been flagged during screening or who present with symptoms. This involves a detailed analysis of clinical data, requiring algorithms to be highly accurate and reliable in differentiating glaucoma from other conditions that may present similarly. Determining an official diagnosis of glaucoma is a more difficult application for AI than screening for suspected disease, and this is not yet established or accepted in many clinical practices. Despite these challenges, there has been exponential growth in research in AI applications for glaucoma diagnosis in the past decade. Most applications focus on OCT, visual fields, or hybrid models that combine structural and functional data.

### 4.1. Leveraging OCT for Glaucoma Diagnosis

OCT, which provides a three-dimensional view of the retina and optic nerve head, is the most widespread tool used to measure structural damage from glaucoma. Table 1 summarizes studies utilizing OCT technology for glaucoma detection, highlighting the diversity of CNN architectures and their respective AUC values. Key findings include high AUC scores ranging from 0.78 to 0.99, underscoring the effectiveness of these models in differentiating glaucoma eyes from normal eyes and in predicting RNFL thickness and different glaucoma stages. In the clinic, structures of interest are automatically segmented by the machine’s software to generate relevant quantitative measures, such as RNFL thickness. Early studies in the 2000s applied machine learning classifiers to time-domain OCT (TD-OCT), showing comparable or better glaucoma detection accuracy than standard OCT parameters alone [35]. With the advent of spectral-domain OCT (SD-OCT) in the 2010s, newer parameters like RNFL thickness enabled sensitivity of 50–80% and specificity of 80–95% for glaucoma diagnosis when analyzed by classifiers [36]. Recently, swept-source OCT (SS-OCT) with scanning speeds of 100,000 A-scans/second has shown potential for earlier glaucoma detection, with algorithms applied to SS-OCT achieving an AUC of 0.95 [37].

Different types of OCT images have been used to develop deep learning algorithms for glaucoma diagnosis, including the OCT conventional report, 2D B scans, 3D volumetric scans, anterior segment OCTs, and OCT-angiography (OCT-A) images. Deep learning models trained with images extracted from the OCT single report can achieve high accuracy in detection of glaucoma [38,39,40,41]. Other models rely on raw OCT scans for model training, rather than previously defined features from automated segmentation software. The usage of raw scans can help to reduce the effects of segmentation error, which can be present in 19.9% to 46.3% of SD-OCT scans [42]. Mariottoni et al. [43] trained a deep learning algorithm to predict RNFL thickness from raw OCT B-scans. These segmentation-free predictions were highly correlated with the actual RNFL thickness (r = 0.983, *p* < 0.001), with a mean absolute error of 2 μm in images of good quality. Thompson et al. [44] also used OCT B-scans to develop a deep learning algorithm that discriminated glaucomatous from healthy eyes. The diagnostic performance of this algorithm was better than using conventional RNFL thickness (AUROC 0.96 vs. 0.87 for the global peripapillary RNFL thickness, *p* < 0.001). OCT volumetric scans of the optic nerve head can provide more comprehensive features and aid in glaucoma detection. Maetschke et al. [45] developed a 3D deep learning model using volumetric OCT scans of the optic nerve head, which achieved a higher AUROC compared to a classic machine learning method using segmentation-based features (AUROC 0.94 vs. 0.89, *p* < 0.05). 

AI-based image analysis of anterior segment OCTs and OCT-A has not yet been explored in depth but does hold potential [46]. Anterior segment OCTs, used to diagnose narrow angles or angle closures, have difficulties related to subjective interpretation. Fu et al. [47] developed a deep learning system trained to detect angle closure from Visante OCT images, which achieved an AUROC of 0.96, sensitivity of 0.90 ± 0.02, and specificity of 0.92 ± 0.008, compared to clinician gradings of the same images. Xu et al. [48] developed a model that could detect gonioscopic angle closure, with an AUROC of 0.928 in the test dataset of Chinese-American eyes and an AUC of 0.933 on the cross-validation dataset, also with AUCs of 0.964 and 0.952 for detecting primary angle closure disease (PACD) based on 2- and 3-quadrant definitions, respectively. OCT-A provides dynamic imaging to map the red blood cell movement over time at a given cross-section. Bowd et al. [49] trained a deep learning model on *en face* 4.5 × 4.5 mm radial peripapillary capillary OCT-A optic nerve head vessel density images. The model showed improvement compared to the gradient boosting classifier analysis of the built-in software in the OCT-A device.

In addition, Machine-to-Machine (M2M) approaches that predict RNFL thickness from fundus photographs are also a growing area of research. OCT has become the standard of care to objectively quantify structural damage in glaucoma [50], but it is expensive and not easily portable. M2M approaches can be used to quantify (not just qualify) glaucomatous damage, especially in low-resource settings without OCT access. Medeiros et al. [51] developed a machine learning classifier for glaucomatous damage in fundus photos, using OCT-derived RNFL thickness as a reference. The model showed a strong correlation (r = 0.832) with actual RNFL values and identified glaucomatous damage with an AUC of 0.944, though 30% of OCT variance was unaccounted for. Thompson et al. [52] employed a similar approach but used a different reference standard from OCT: the Bruch’s membrane opening-minimum rim width (BMO-MRW) parameter. Again, predictions from the deep learning model were well correlated with the actual BMO-MRW values (Pearson’s r = 0.88, *p* < 0.001), with an AUC of 0.933 for distinguishing deep learning predictions from glaucomatous and healthy eyes.

As we venture into the realm of AI’s practical applications in glaucoma diagnosis, it is crucial to shift our focus from controlled research environments to real-world clinical settings. The efficacy and reliability of AI technologies must be critically evaluated in diverse clinical environments to understand their performance and applicability in routine clinical practice. The intricacies of real-world application, such as varied patient demographics, differing equipment, and non-standardized operating procedures, present unique challenges that are not typically encountered in controlled research settings.

Recent studies have begun to address this gap by conducting field trials and observational studies in various clinical settings. For example, the use of AI in community eye clinics and in regions with limited access to specialized care provides valuable insights into the performance of these technologies outside traditional research environments. These studies often highlight the need for robust AI models that can adapt to varying image qualities and different patient populations. Additionally, the integration of AI into existing healthcare workflows and its impact on clinical decision-making processes are being actively explored. These real-world evaluations are critical in ensuring that AI technologies not only meet the stringent requirements of clinical validation but also demonstrate practical utility and scalability in diverse healthcare settings.

**Table 1 bioengineering-11-00122-t001:** Summary of studies using OCT technology to detect glaucoma.

Author	Year	Ref.	Architecture	AUC	Measurements
Glaucoma Detection					
Muhammad et al.	2017	[41]	Custom HDLM	-	Healthy/suspect eyes and eyes with mild glaucoma
Asaoka et al.	2019	[53]	Novel CNN	0.937	Early POAG vs. no POAG
Maetschke et al.	2019	[45]	3D CNN	0.94	POAG vs. no POAG
Akter et al.	2023	[54]	Hybrid CNN combining SqueezeNet, ResNet18, and VGG16	0.988	Glaucoma vs. normal eyes
He et al.	2023	[55]	Custom Transformer Network	0.9999	Retinal disease types
Christopher et al.	2023	[56]	ViT	0.80	Glaucoma patients with vs. without surgery
Angle Closure Detection					
Fu et al.	2019	[47]	VGG-16 with Transfer Learning	0.96	Detection of angle closure
Fu et al.	2019	[57]	Multilevel Deep Network	Visante AS-OCT dataset: 0.9619 Cirrus HD-OCT dataset: 0.9524	Open vs. Closure angle
Xu et al.	2019	[48]	ResNet18	detecting gonioscopic angle closure: cross-validation dataset: 0.933 test dataset: 0.928 detecting PACD: cross-validation dataset: 0.964 test dataset: 0.952	Open vs. Closure angle
Hao et al.	2019	[58]	Custom MSRCNN	0.9143	Open angle vs. narrowed angle vs. closed angle or synechia
Randhawa et al.	2023	[59]	Custom CNN	CHES: 0.917 Singapore: 0.894 USC: 0.922	Detection of gonioscopic angle closure
Tissue Segmentation and Prediction					
Devalla et al.	2018	[60]	Custom CNN for RNFL analysis	-	Digitally staining six tissue layers of ONH
Thompson et al.	2019	[52]	ResNet34	0.933	Global BMO-MRW prediction
Medeiros et al.	2019	[51]	ResNet34	0.944	RNFL thickness prediction
Jammal et al.	2020	[61]	M2M Network	0.529	RNFL prediction
Lee et al.	2020	[62]	NASNet	0.990	GCIPL and RNFL
Lee et al.	2021	[63]	M2M	-	RNFL prediction
Medeiros et al.	2021	[64]	CNN	-	Detection of RNFL thinning
Hood et al.	2022	[39]	CNN	-	RNFL probability maps
Bowd et al.	2022	[49]	VGG16-CNN	0.97	RNFL thinning for healthy vs. glaucoma eyes
Outcome Improvement and Management					
Wang et al.	2020	[65]	CNN for Multi-Task Learning	HK dataset: 0.977 Stanford dataset: 0.933	Yes vs. No glaucoma
Russakoff et al.	2020	[66]	Custom gNet3D-CNN	gNet3D with homogenization: 0.88 gNet3D without homogenization: 0.82	Referable vs. non-referable glaucoma

### 4.2. Visual Fields and the Power of Hybrid Models

In addition to OCT scans, visual fields have been explored for AI-enabled diagnosis of glaucoma. Standard automated perimetry (SAP) using the Humphrey Field Analyzer has been the main method for assessing visual field defects in glaucoma. SAP provides numerical data on light sensitivity at different visual field locations, as well as summary indices like mean deviation. Beginning in the 1990s, machine learning techniques like artificial neural networks were applied to analyze and interpret SAP visual fields for glaucoma diagnosis. More recently, as we mentioned previously, CNNs have also been trained using raw visual field data or probability maps to classify fields as normal versus glaucomatous [53,67].

Li et al. [67] trained a deep learning algorithm with the probability map of the pattern deviation image, showing that it had superior performance in distinguishing normal from glaucomatous visual fields (accuracy 87.6%) than either human graders (62.6%), the Glaucoma System 2 (52.3%), or the Advanced Glaucoma Intervention Study criteria (45.9%). Although it is considered one of the most robust algorithms using visual fields, one limitation is that the input pattern deviation images may preclude early glaucoma from being identified. Elze et al. [68] used “archetypal analysis” to classify patterns of visual field loss, such as arcuate defects, finding good correspondence to human classifications from the Ocular Hypertension Treatment Study (OHTS). With a follow-up study also using archetypal analysis, Wang et al. [69] classified central visual field patterns in glaucoma, showing that specific subtypes with nasal defects were associated with more severe total central loss in the future. Brusini et al. [70] developed a model that could identify local patterns of visual field loss and classify and quantify the degree of severity based on subjective assessments. Li et al. [71] developed iGlaucoma mobile software, which is a smartphone application-based deep learning algorithm that extracts data points in the visual field using optical character recognition techniques. This software outperformed ophthalmologist readers and has undergone real-world prospective external validation testing. 

Early studies suggest that hybrid deep learning models that combine structural and functional tests have increased performance over models trained with either test alone [72]. Such models better mimic a clinical diagnosis from eye specialists, which is typically multimodal and does not rely on a single imaging modality as input. Xiong et al. [73] showed that a multimodal algorithm using both visual fields and OCT scans to detect glaucomatous optic neuropathy had superior performance compared with models that relied on each modality alone. Other groups have focused on prediction models, such as prediction of visual field sensitivities from RNFL thickness from OCT [74,75,76,77]. Using fundus photographs to predict RNFL thickness has been shown to predict future development of field defects in eyes of glaucoma suspects [63]. Lee et al. [78] trained a deep learning algorithm to predict mean deviation from optic disc photographs, which could be useful when SAPs are not available. Sedai et al. [79] combined multimodal information into a model, using clinical data (age, IOP, inter-visit interval), circumpapillary (cp) RNFL thickness from OCT, and visual field sensitivities to predict cpRNFL thickness at the subsequent visit. This model showed consistent performance among suspects and cases and could potentially be used to personalize the frequency of follow-up visits for patients. 

### 4.3. Challenges and Future Prospects for AI in Glaucoma Diagnosis

One major barrier shared by all algorithms trained to diagnose glaucoma is the lack of a gold standard definition for the presence and progression of this disease. Numerous studies cite high interprovider variability in glaucoma diagnosis [80,81], which serves as the reference standard for evaluating algorithm outputs. A clear, concrete definition of glaucoma could help set the bar for model accuracy [34]. For example, diabetic retinopathy has an agreed upon classification system, allowing a more straightforward approach to developing AI for diagnostic applications with proven success; in 2018, a deep learning system for diagnosis of this disease in diabetic patients received FDA approval for use in primary care clinics [20]. This system documents the appearance of the optic nerve but is not approved to diagnose glaucoma at this time.

Another key challenge in diagnosis is how to design the interface between the clinician and the AI model. For instance, recent work has demonstrated that professional radiologists selectively comply with AI recommendations in a suboptimal way, which can lead to worse performance than desired [82]. Relatedly, selective compliance has also led to issues with racial bias in other domains [83]. Techniques such as explainability have been proposed to help bridge this gap, though significant challenges remain to ensuring reliability of explainability techniques [84]. As an alternative, high-quality uncertainty quantification has been shown to help improve end-user trust in other domains [85] and may be valuable for clinical decision support systems as well.

Moreover, each imaging modality also poses its own set of challenges to implementation in real-world settings. OCT machines are expensive and therefore not as applicable for low-resource areas. Additionally, like fundus images, anatomic abnormalities can influence results, and there is a lack of interchangeability across OCT devices [46]. Visual field testing is subjective and can be affected by patient factors such as attention and fatigue [86]. Additionally, most models are typically trained using visual field tests labelled as reliable and may not be able to identify unreliable exams, which are very common in clinical settings. Finally, because structural changes are known to precede functional damage in glaucoma, it can be difficult to provide an early diagnosis using visual fields. As a result, many AI applications using visual fields are better suited to assess disease progression, rather than diagnosis. There are also several barriers to the development and implementation of hybrid models. Such models require paired data from imaging modalities in training and testing datasets, which imposes limits on the availability and feasibility of data collection. When using multiple input types, there is also a need to add more training data to avoid overfitting [87]. 

## 5. Predictive Power: AI’s Forecasting Capabilities in Glaucoma

In the domain of glaucoma management, predictive models fortified by AI stand as powerful tools, offering insights into a patient’s risk and the potential speed of disease progression [88]. These models leverage cutting-edge machine learning techniques, such as regression analyses, neural networks, and ensemble methods, to distill vast arrays of patient data into actionable predictions. For instance, by analyzing a patient’s genetic makeup, historical IOP trends, and imaging data, AI can forecast the likelihood of rapid disease progression.

### 5.1. Forecasting Glaucoma Development

Delving deeper into the predictive realm, forecasting glaucoma to identify those with future disease development remains an understudied area. Thakur et al. [89] used deep learning from more than 60,000 fundus photographs from OHTS to predict glaucoma prior to clinical signs. This study showed consistent performance in predicting glaucoma development prior to disease diagnosis, with AUCs of 0.88 for 1 to 3 years before glaucoma and with an AUC of 0.95 for glaucoma diagnosis after onset [89]. Yoon et al. [88] studied the oral microbiome in 96 glaucoma patients and 25 controls, identifying the genus Lactococcus as a key predictor of glaucoma. Regression models linked Lactococcus, Candidatus Pelagibacter, and Atopobium to glaucoma severity. Their findings suggest that oral microbiome imbalances could assist in diagnosing and managing glaucoma. Fei Li’s group [90] introduced deep learning models, DiagnoseNet and PredictNet, to predict glaucoma diagnosis and progression using retinal fundus photographs. Trained on data from 14,905 individuals, PredictNet identified high-risk patients with an AUC of 0.91, distinguishing progression rates between risk groups. These models, with further validation, could revolutionize early glaucoma intervention and management. 

### 5.2. Monitoring Disease Progression: AI’s Adaptive Evolution

Having explored the remarkable predictive power of AI in glaucoma diagnosis, it becomes evident that these capabilities can be extended to another crucial aspect of glaucoma management: the detection and monitoring of disease progression. The same advanced algorithms that predict the likelihood of glaucoma development are also instrumental in tracking subtle changes over time, offering a seamless shift from prediction to proactive management. This transition to monitoring disease progression highlights AI’s versatility in providing continuous, comprehensive care for glaucoma patients.

Table 2 provides a comprehensive summary of pivotal studies that have employed AI methods to detect progression in glaucoma. These studies span over two decades, showcasing the evolution of AI from rudimentary supervised machine learning techniques to more sophisticated unsupervised methods and Bayesian hierarchical models. The table enumerates distinct research efforts, detailing the number of eyes or images analyzed, the instruments used, and the AI approach taken. This progression underscores AI’s adaptability in handling different data forms and complexities, reflecting a trend from supervised machine learning, which relies on labeled data, to unsupervised learning, which discovers patterns in the data without pre-labeled outcomes. For instance, early works by Brigatti et al. [91] and Lin et al. [92] demonstrated the potential of supervised machine learning with SAP data. As the field advanced, Medeiros et al. [93] and others integrated multiple data sources, such as combining SAP with SLP, to enhance the predictive power of their models. Unsupervised machine learning became more prevalent in later studies, as evidenced by the work of Goldbaum et al. [94], which allowed for a more nuanced analysis of large datasets, identifying subtle patterns indicative of disease progression. In 2023, the trend in glaucoma progression detection using AI continued to evolve, with studies like Mariottoni et al. [95] and Christopher et al. [96] leveraging supervised machine learning on large datasets from SD-OCT and OCT scans. Hussain et al. [97] uniquely combined both supervised and unsupervised machine learning approaches, reflecting a growing trend towards hybrid methodologies that capitalize on the strengths of both techniques to achieve more accurate and robust predictions.

The long-term monitoring and follow-up of AI-based interventions in glaucoma care are critical to assess their sustainability and efficacy over extended periods. Continuous evaluation of AI models in clinical practice is essential to ensure their reliability and accuracy in diagnosing and predicting glaucoma progression. Longitudinal studies are needed to track the performance of these AI systems, analyzing their impact on patient outcomes, treatment adjustments, and disease progression. Furthermore, follow-up studies would provide valuable insights into the adaptability of AI models to evolving clinical data and their capacity to maintain accuracy over time. Such ongoing assessments are pivotal in validating the long-term utility of AI in glaucoma management and ensuring that these technologies continue to meet the dynamic needs of patient care.

**Table 2 bioengineering-11-00122-t002:** Summary of studies using AI methods to detect progression in glaucoma.

Author	Year	Ref.	No. of Eyes/Images	Instrument	Approach
Brigatti et al.	1997	[91]	233 visual fields from 181 patients	Octopus G1	Back propagation NN
Lin et al.	2003	[92]	80 patients	SAP	Back propagation NN
Sample et al.	2005	[98]	191 patient eyes	SAP	vB-ICA-mm
Goldbaum et al.	2005	[94]	189 normal eyes and 156 eyes diagnosed with GON	SAP	vB-ICA-mm
Medeiros et al.	2011	[93]	434 eyes of 257 participants	SAP and SLP	Bayesian hierarchical model
Medeiros et al.	2012	[99]	711 eyes from 357 glaucoma patients or suspects	SAP	Bayesian hierarchical model
Bowd et al.	2012	[100]	264 eyes of 193 participants	SAP and CSLO	RVM
Medeiros et al.	2012	[101]	242 eyes from 179 patients	SAP and CSLO	Conventional approach + Bayesian hierarchical model
Medeiros et al.	2012	[102]	352 eyes from 250 patients	SAP	Conventional approach + Bayesian hierarchical model
Yousefi et al.	2013	[103]	107 progressing and 73 stable glaucoma	SAP	MLCs
Murata et al.	2014	[104]	5049 (training data) and 911 (test data)	SAP	VBLR
Belghith et al.	2014	[105]	36 eyes processing, 210 eyes non-processing	HRT	Markov Random Field + VEM
Belghith et al.	2015	[106]	117 eyes from 75 participants	SD-OCT	Bayesian framework with SVDD classifiers
Yousefi et al.	2016	[107]	859 abnormal SAP and 1117 normal SAP	SAP	GEM + VIM
Yousefi et al.	2018	[108]	2085 eyes of 1214 subjects	SAP	ML-based index
Wang et al.	2019	[109]	12,217 eyes from 7360 patients	SAP	Unsupervised ML
Jammal et al.	2022	[110]	7501 eyes of 3976 subjects or suspected glaucoma	SD-OCT	Linear mixed models
Hu et al.	2022	[111]	4512 glaucoma patients	EHRs	NLP
Mariottoni et al.	2023	[95]	14,034 SD-OCT scans from 816 eyes of 462 individuals	SD-OCT	CNN
Hussain et al.	2023	[97]	105 eyes (reduced to 86 eyes)	OCT	CNN + LSTM
Christopher et al.	2023	[96]	3327 scans from 1096 eyes of 550 patients	OCT	Supervised ML
Hou et al.	2023	[112]	4211 eyes from 2666 patients	OCT	GTN
Tian et al.	2023	[113]	5167 patients	OCT	ViT

### 5.3. Vision Transformers: A New Frontier

Continuing this trajectory, recent advancements have seen the application of vision transformers, an AI approach that has been in existence for some time, in the specific context of glaucoma detection and progression. This adaptation of established AI technology to glaucoma research, particularly in the years 2022 and 2023, highlights its growing relevance and potential in this specialized field. For example, a study by Hu and Wang [111] utilized massive transformer-based language models applied to clinical notes from Electronic Health Records (EHRs) to predict glaucoma progression requiring surgery, showcasing the potential of natural language processing in glaucoma research. Similarly, Hu et al. [114] introduced GLIM-Net, a novel transformer-based network for forecasting chronic glaucoma from irregularly sampled sequential fundus images, demonstrating the transformer architecture’s efficacy in medical imaging, particularly in capturing the temporal progression of glaucoma. Luo et al. [115] further advanced the field by developing a generalization-reinforced semi-supervised learning model for glaucoma detection and progression forecasting, introducing the comprehensive Harvard Glaucoma Detection and Progression Dataset to address the challenge of limited labeled data. Studies like those by Hou et al. [112] and Tian et al. [113] have further extended the use of transformer networks to analyze longitudinal OCT data, demonstrating the versatility and efficacy of these models in handling diverse data types. These transformer models, known for their proficiency in capturing complex dependencies in data, have demonstrated remarkable capabilities in predicting glaucoma progression, outperforming traditional machine learning models in accuracy and predictive power. The utilization of transformers signifies a pivotal shift towards more advanced AI techniques that can efficiently process and interpret vast and intricate datasets, offering a promising direction for future glaucoma research and clinical decision making.

## 6. AI Assistance for Precision Medicine and Personalized Treatment in Glaucoma

In the evolving landscape of medical care, precision medicine emerges as a beacon, moving away from the ‘one-size-fits-all’ approach to a more tailored strategy, addressing the unique genetic, environmental, and lifestyle factors of individual patients [116]. Glaucoma, with its diverse manifestations and progression rates, stands to benefit immensely from such a tailored approach [117]. Given its insidious progression and the potential for irreversible visual impairment, it is paramount to ensure that treatment strategies align closely with individual patient needs. AI, with its vast data processing capabilities and predictive modeling, could play a pivotal role in harnessing the power of precision medicine for glaucoma [118]. By analyzing patient details and disease characteristics, AI can help to provide more personalized treatment plans, promising improved outcomes and patient-centric care [118].

Precision medicine aims to provide the right treatment to the right patient at the right time. However, it is a challenging process to apply precision medicine in glaucoma. Key requisites include amalgamating disconnected health data, uncovering predictive patterns, and integrating insights at the point of care. AI models that can address these needs are emerging as clinical decision support tools. Rigorous prospective testing is still needed, given limitations of retrospective datasets. Evaluating real-world efficacy, cost-effectiveness and ethical factors related to AI-guided treatment recommendations should be a priority [119]. Ongoing physician oversight and vigilance are thus essential, even with automated aids.

While the potential of AI in revolutionizing glaucoma treatment is undeniable, it is not without its set of challenges and considerations. Technically, the efficacy of AI is bound to the quality of the data it processes; hence, issues with data integrity or incompleteness can hamper accurate predictions [6]. Additionally, biases inherent in algorithms, often stemming from non-representative training data, can lead to skewed or even erroneous outcomes [120]. Proper model validation becomes crucial to ensure that AI predictions stand up to real-world scrutiny. On the ethical front, concerns about transparency in AI decision making, ensuring fairness across diverse patient populations, and upholding patient autonomy in treatment decisions loom large [121]. Clinically, seamlessly integrating AI tools into established workflows can be daunting, compounded by varying levels of clinician acceptance and trust in these systems. As AI’s footprint in glaucoma care expands, addressing these multifaceted challenges becomes paramount to fully harnessing its transformative potential, while safeguarding patient interests and trust.

The integration of AI technologies in clinical practice brings to the forefront the issue of cost-effectiveness, particularly in low-resource settings where glaucoma is prevalent. It is critical to evaluate the cost–benefit analysis of AI applications in these settings, considering the initial investment in technology versus long-term healthcare savings. AI can potentially reduce the need for frequent and extensive diagnostic testing, thus lowering overall healthcare costs. However, the initial costs for infrastructure, training, and implementation of AI systems can be substantial, especially in resource-limited settings.

Moreover, the accessibility of AI-based glaucoma care in these environments is a paramount concern. The development and implementation of low-cost, scalable AI solutions that can be seamlessly integrated into existing healthcare systems are essential. For instance, AI-enabled telemedicine initiatives could extend expert diagnostic capabilities to remote areas, enhancing access to specialized care. Also, the automation of routine diagnostic procedures by AI could alleviate the burden on healthcare professionals in these settings. In summary, the exploration of cost-effective and accessible AI solutions is vital for the equitable distribution of healthcare resources and the effective management of glaucoma globally.

## 7. The Role of AI in Improving Glaucoma Outcomes

AI has ushered in a new era of possibilities in medical research and preoperative surgical decision making. In the context of glaucoma treatment, AI harnesses its computational capabilities to offer a new approach to choosing the best surgical intervention. For example, AI has been used to compare glaucoma surgery techniques for a given glaucoma diagnosis by evaluating a large number of clinical and anatomic parameters that were previously difficult to assess. In a study using the DD-SIMCA method and machine learning to assess various clinical parameters, lens extraction was found more effective than laser iridotomy in treating primary angle-closure glaucoma (PACG), leading to improved anterior chamber topography and reduced IOP [122]. In the future, AI could be used to help generate valuable comparisons between surgical techniques, affecting clinical treatment decisions.

A particularly promising use of AI in medicine has involved the application of electronic medical record (EMR) data to personalize patient intervention. Machine learning algorithms have been trained to model complex relationships such as risk factors, clinical exam findings, and imaging results to predict specific and measurable outcomes, e.g., quality of life, morbidity, and mortality. Many areas of surgery, including cataract surgery, esophageal cancer surgery, and deep brain stimulation surgery, have used this machine learning modeling to predict surgical results [123,124]. 

Qidwai et al. [125] recently developed the Adaptive Neuro-Fuzzy Inference System (ANFIS), an AI algorithm that considers multiple baseline clinical parameters to suggest a specific minimally invasive glaucoma surgery (MIGS) treatment. They utilized a retrospective case series of patients who underwent one of four MIGS procedures to build a predictive model for the MIGS procedure using several baseline clinical characteristics. In the near future, as the number of interventions for glaucoma increases, AI programs that can leverage patient information to recommend the most appropriate treatment option could greatly impact the field and improve patient outcomes. 

In vitreoretinal surgery, Nespolo et al. [126] utilized 606 surgical image frames to train a model known as YOLACT++, which is an instance of a fully convolutional neural network specifically designed for segmentation tasks. They have also built a platform where this AI system could locate, classify, and segment tissue and instruments in real time. There were two proposed benefits: This system could recognize unintended instrument usage during surgical guidance and could analyze surgical instrument movements postoperatively for vitreoretinal surgeon training. In cataract surgery, there has been research using AI in real time to track eye structures, such as the pupil [127] and surgical phases [127,128]. Computer vision tools used to track surgery steps, instruments, tissues, and structures could improve the training of surgeons [129] and outcomes of cataract surgery by improving rhexis symmetry and providing feedback during potentially harmful instrument movements [127].

Besides providing intraoperative guidance, AI can assess a surgeon’s performance, potentially improving surgical techniques and minimizing surgical complications. One method is by retrospectively analyzing surgery through automated processing of surgical recordings to identify the steps and to track surgical patterns, which may even allow for a streamlined postoperative evaluation [129,130]. Soon, there could be integrations between AI-related intraoperative tracking and resident surgical training. By providing novice surgeons with real-time intraoperative feedback or detailed postoperative feedback, programs could allow trainees to review surgical patterns as a way to improve their surgical technique. In addition, AI could potentially be useful for residency training programs, hospitals, and individual clinics to develop metrics for surgeons and a means to track overall resident performance over time [129]. This application of AI has yet to be seen in glaucoma surgery or training. However, the potential for AI to minimize surgical errors and to improve training is promising.

Cost et al. [131] combined intraoperative OCT’s superior visualization with deep learning to enhance the accuracy of graft orientation identification during Descemet membrane endothelial keratoplasty (DMEK). Their method involved iOCT image segmentation using AI, conversion to a one-pixel line, and assessment of graft orientation based on its curling behavior. While iOCT is emerging in some MIGS procedures for glaucoma treatment, the integration of AI tools remains largely unexplored, offering potential for enhanced surgical precision. Lin et al. [132] designed a real-time CNN model to pinpoint the trabecular meshwork in gonioscopy videos, using 378 gonioscopy images for training. While not yet tested in real-time clinical settings, this model holds potential for enhancing surgical training and aiding novice ophthalmologists during procedures.

Moreover, the incorporation of AI in glaucoma care provides a unique opportunity to tailor patient-centered approaches. AI’s adaptability allows for the customization of diagnostic and treatment strategies, taking into account individual patient characteristics, preferences, and experiences. For example, AI can analyze patient feedback and outcomes to refine treatment plans, offering personalized care paths that align with each patient’s unique health profile. Furthermore, AI-enabled platforms can facilitate patient engagement and education, empowering patients to actively participate in their care process. This personalized approach, driven by AI, not only enhances patient satisfaction and compliance but also optimizes clinical outcomes, making it a crucial aspect of future glaucoma care strategies.

In conclusion, AI’s transformative impact on ophthalmology is palpable, with applications spanning localization, training, visualization, and surgical precision. While glaucoma surgery is poised to benefit from these advances, the potential for AI to minimize errors, enhance surgical skills, and improve patient outcomes across ophthalmic surgeries is an exciting frontier awaiting further exploration.

## 8. AI for Patient Education, Counseling, and Improving Medication Adherence

One area of interest in AI is its potential for improving glaucoma medication adherence, patient education, and counseling. Glaucoma requires diligent and consistent medication management to prevent vision loss [133,134,135]. As shown in multiple landmark clinical trials, IOP is currently the major proven modifiable risk factor in glaucoma [133,134]. Therefore, one of the most supported interventions to manage glaucoma is using medicated eyedrops to lower IOP. However, medication adherence is a significant challenge for many patients, with reported rates of glaucoma medication non-adherence up to 80% [136]. 

Given the importance of medication adherence in glaucoma management, numerous studies have examined patient behaviors and identified barriers to effective medication adherence, alongside experimenting with various interventions [137,138]. In the literature, possible barriers to glaucoma medication adherence include poor glaucoma education, forgetfulness, difficulties with medication administration, cost, schedule, and skepticism that glaucoma medications are effective and that glaucoma will lead to vision loss [139,140,141]. In response to these challenges, AI technologies are emerging as potential solutions. AI-powered systems can provide personalized education and reminders, tailored to individual patient needs and behaviors, thereby enhancing understanding and adherence. For instance, AI algorithms can analyze patient data to predict when individuals are most likely to forget doses and to send timely reminders. Additionally, AI-driven applications could assist in simplifying medication schedules or offering virtual assistance for correct drop administration, making the process more manageable for patients. Research into the frequency of these solutions found that low self-efficacy, forgetfulness, and difficulty with drop administration and medication schedules were all highly associated with poor glaucoma medication adherence [142]. The integration of AI into glaucoma care not only promises to improve adherence rates but also opens avenues for more personalized and effective patient management strategies. While extensive literature exists on the examination of glaucoma medication compliance, there has been limited exploration of AI-related research to improve adherence. One noteworthy study that employed AI to facilitate glaucoma medication adherence was led by a team at the University of Michigan, who created an AI-powered bottle sleeve to detect eye drop usage, measure fluid levels, and relay this data to healthcare professionals [143]. Using a machine learning algorithm, they accurately monitored medication use, outperforming a rule-based method that relied on factors like bottle orientation. This highlights the potential of machine learning to enhance new technologies in medication adherence tracking. Integrating AI with technologies like sensors can bolster reliability and provide a deeper understanding of patient compliance factors [143,144]. 

## 9. AI in Glaucoma: Future and Challenge

The application of AI in glaucoma decision making has seen significant growth over the past decade, especially in areas like OCT, visual fields, and hybrid models that merge structural and functional data. Validating these AI models is crucial; their reliability and clinical relevance are determined by comparing AI predictions with real-world patient outcomes. Such validations ensure the model’s accuracy, facilitating its integration into clinical workflows and aiding clinicians in making informed decisions. 

AI and machine learning techniques show promise in automating continuous disease monitoring, synthesizing assessments from multiple algorithms, and integrating multimodal data. Additionally, innovations like AI-powered chatbots and smartphone applications are emerging to assist visually impaired patients with medication adherence, showcasing the potential of AI to revolutionize glaucoma care. For aging patients, the ability to take care of their own medications diminishes due to a number of age-related issues, including reduced visual acuity, challenged fine motor skills, and memory deficits [145]. Because of these difficulties, there has been interest in how AI-powered chatbots could aid medication understanding and adherence in older adults [145]. Although AI-powered chatbots have not been studied for glaucoma patients with impaired eyesight, Tran et al. [146] developed an app integrated with AI software that is able to identify glaucoma medication bottles for visually impaired patients.

Interdisciplinary collaboration is vital in the development and implementation of AI systems for glaucoma. Combining the expertise of clinicians, data scientists, ethicists, and patients ensures a holistic approach to AI development, fostering innovations that are clinically relevant, ethically sound, and patient-centric. Clinicians provide essential insights into clinical needs and practicalities, while data scientists contribute technical expertise in AI and machine learning. Ethicists ensure that these technologies are developed and used in a manner that upholds ethical standards and patient rights. Most importantly, involving patients in the development process ensures that the AI systems are tailored to meet their specific needs and preferences, enhancing the overall effectiveness and acceptance of these technologies. This interdisciplinary approach is not just beneficial but necessary for the successful integration of AI in glaucoma care.

The effectiveness of AI in glaucoma management heavily relies on the diversity and representativeness of the datasets used for training and validating these systems. Currently, there is a critical need to ensure that datasets encompass a wide range of demographic variables, including age, ethnicity, and genetic backgrounds, as well as diverse clinical presentations of glaucoma. In particular, most datasets include images or data from European or Asian individuals, with low representation from African ancestry individuals. The lack of diversity in training datasets may result in worse performance of the AI model in new data from a different population [46]. AI models are known to reflect biases inherent in the training dataset, with accuracy affected by variables such as differences in optic disc size or fundus pigmentation [147]. As glaucoma prevalence and response to treatment can vary significantly across different populations, incorporating broad and inclusive datasets in AI research is crucial for equitable and effective glaucoma care. By prioritizing dataset diversity, researchers and clinicians can enhance the generalizability and reliability of AI applications in glaucoma, ensuring that advances in AI-driven healthcare benefit a global and diverse patient population. 

In addition to ensuring that training datasets include more diverse populations, there is also a need to study approaches to effectively mitigate bias from existing AI models. For example, studies could evaluate different techniques to train models on existing datasets and assess whether these transfer successfully to overly affected African-ancestry individuals. Besides training models using more diverse datasets, adapting foundation models to patient populations presents a promising pathway. Foundation models like RETFound and FLAIR, which have been pre-trained on vast amounts of unlabeled retinal images, offer generalizable representations and enable label-efficient model adaptation in various applications [148,149]. RETFound, trained on 1.6 million unlabeled retinal images, has shown superior performance in the diagnosis and prognosis of sight-threatening eye diseases with fewer labeled data points [148]. Similarly, FLAIR, a pre-trained vision-language model, integrates expert knowledge through descriptive textual prompts and demonstrates strong generalization capabilities, especially under domain shifts or unseen categories [149]. These foundation models, by learning useful features during pre-training stages, are expected to adapt to new patient populations with much fewer data points, potentially alleviating the annotation workload of experts and enabling broad clinical AI applications.

Another significant challenge in AI’s application to glaucoma is the absence of a universally accepted definition for the disease’s diagnosis and progression. High interprovider variability in glaucoma diagnosis complicates the evaluation of algorithm outputs. Each imaging modality, whether OCT or visual fields, comes with its unique set of challenges, from high costs to subjectivity influenced by patient factors. Hybrid models, which require paired data from different imaging modalities, face barriers in data collection and the risk of overfitting. Furthermore, as glaucoma severity intensifies, patients often struggle with medication adherence due to deteriorating vision, emphasizing the need for tools like AI-powered smartphone apps. However, as new tools emerge, research must also focus on assessing patient engagement, treatment adherence, and follow-up to ensure that these innovations truly benefit the patients they aim to serve.

In addition to the technical and ethical challenges already discussed, the clinical adoption of AI in glaucoma treatment faces several crucial hurdles. These include regulatory complexities surrounding AI applications in healthcare, which require rigorous scrutiny to ensure patient safety and efficacy. Issues of patient privacy and data security are paramount, especially considering the sensitive nature of medical data. The question of data ownership, particularly in the context of patient-generated data, raises significant legal and ethical considerations. Moreover, the matter of liability in cases of misdiagnosis or treatment errors involving AI systems remains an area of active debate and legal evolution. Addressing these challenges is essential for the responsible and effective integration of AI in clinical settings.

In conclusion, the future of AI in glaucoma care hinges on overcoming challenges related to dataset diversity, algorithmic bias, and the standardization of disease definitions. By addressing these issues, AI can significantly enhance glaucoma diagnosis, monitoring, and management, offering more personalized and effective care for a globally diverse patient population. Continued research and development in this field promises to not only improve patient outcomes but also to revolutionize the approach to managing this complex and prevalent eye disease.

## Figures and Tables

**Figure 1 bioengineering-11-00122-f001:**
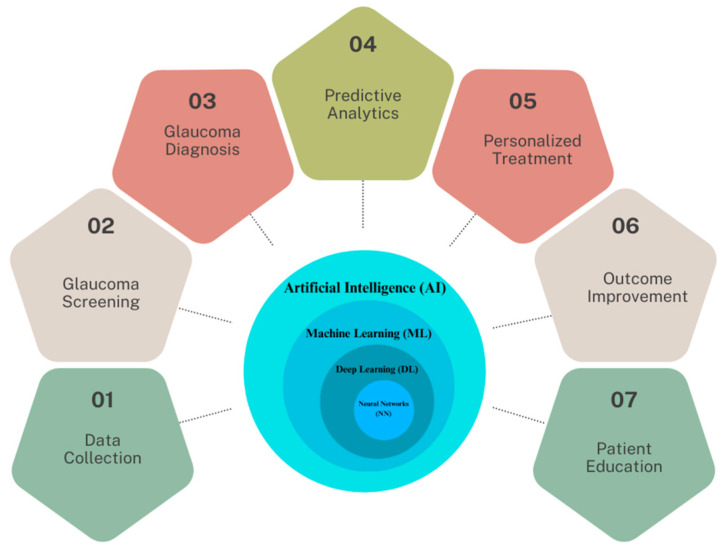
The spectrum of AI applications in glaucoma healthcare.

## Data Availability

Not applicable.

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
