# Peer review of "Advancing Glaucoma Care: Integrating Artificial Intelligence in Diagnosis, Management, and Progression Detection"

_bioengineering, 2024, doi:10.3390/bioengineering11020122_

Round 1

Reviewer 1 Report

Comments and Suggestions for Authors

The authors provide a review of the use of AI in clinical glaucoma care. The authors cover many important topics related to AI adoption in glaucoma care and they extensively reference the relevant literature covering these topics. However, there are several weaknesses that should be addressed:

-       Fig 1 is somewhat confusing. It references important topics / techniques related to AI uses for glaucoma, but the figure does not provide insight into the relationship between these topics. For instance, there are arrows between AI techniques and “Glaucoma Screening” and between “Data Collection” and “Personalized Treatment.” However, the figure doesn’t illustrate how these topics relate to one another effectively. Another example: it references AI, ML, DL, and NNs, but doesn’t suggest the relationship between them (e.g., NN and DL as a subset of ML, ML as a subset of AI). The pictures / icons used in the figure are also more distracting than helpful.

-       Starting on line 168: “The attention maps from DeiT models focused on localized areas such as the neuroretinal rim, indicating a more detailed and clinically relevant feature analysis.” I’m not entirely sure what “more detailed” means in this context, but I think this is overstating the case a bit. What we can say is that these specific models put some focus on areas that are also used in manual review of images (i.e., the neuroretinal rim). I would restate this.

-       Some aspects of Table 1 are unclear:

o   In the “Architecture” column, some items are unclear / don’t refer to model architectures. For ex, Duvella et al. 2018 is listed as a “Digital Stain of RNFL.” It would be better described as a custom CNN architecture.

o   The table seems to be comparing AUCs for models that are performing very different tasks – segmentation, glaucoma detection, identifying closed angles, surgery prediction. The table should be reformatted / broken up into different tables to make it clear to readers that the AUC values can’t be directly compared for these tasks.

o   Some of the notation is unclear / inconsistent. E.g., What does an AUC of “0.977+0.933” mean?

-       On line 417 “2022 and 2023 have seen the emergence of vision transformers”: ViT have been around a little longer than that, more correct to say they have emerged into use for glaucoma tasks.

-       The authors don’t mention several important challenges to clinical adoption such as regulatory issues, patient privacy / data security, data ownership, liability, etc. I think a detailed discussion of these topics is outside the scope of the manuscript, but they should be mentioned / described briefly.

-       Title should probably be updated since it only mentions diagnosis and detection, but the paper covers ongoing management, progression detection, etc.

-       Minor issues:

o   Uses EHR acronym without introducing it, then later introduces / uses EMR

o   Line 499: “Nespolo et al. [127] used 606 surgical image frames to train an instance segmentation fully CNN named YOLACT++”. Did this mean to describe an instance of a fully convolutional network?

o   Several other typos throughout, could use additional proofreading

Comments on the Quality of English Language

Some additional proofreading for clarity would be helpful.

Author Response

We thank the editor for the insightful comments below, which have helped to 
improve the paper. Please check the pdf attachment.

Reviewer 2 Report

Comments and Suggestions for Authors

Dear authors,

I have now completed the review of the manuscript titled "The Clinical Integration of Artificial Intelligence for Glaucoma Diagnosis and Detection"

The manuscript is interesting and, in general, fair written. Therefore, I would like to suggest a minor revision, but authors may interested in my furthere suggestions:

1. The paper could delve deeper into how the AI models were trained, specifically addressing the diversity of the training data. Glaucoma presentation can vary across different ethnicities and ages; thus, a lack of diverse data can lead to biased AI models with limited applicability.

2. While the paper discusses AI's capabilities, it lacks detailed discussion on the interpretability of these AI systems. For clinical integration, understanding how AI reaches a conclusion is crucial for trust and effective usage by clinicians.

3. The paper could benefit from more emphasis on the clinical validation of these AI technologies in real-world settings. It's important to understand how these technologies perform outside controlled research environments and in diverse clinical settings.

4. The paper could explore the cost-effectiveness of integrating AI into clinical practice and its accessibility, especially in low-resource settings where glaucoma is prevalent but healthcare resources are limited.

5. The paper could discuss the need for long-term monitoring and follow-up studies to assess the sustainability and long-term efficacy of AI-based interventions in glaucoma care.

6. An exploration of how AI can be tailored to enhance patient-centered care, considering patients' perspectives, experiences, and preferences, would add value to the discussion.

7. The paper could highlight the importance of interdisciplinary collaboration in developing and implementing AI systems for glaucoma, involving clinicians, data scientists, ethicists, and patients.

Thank you for your valuable contributions to our field of research. I will wait a revised manuscript.

Author Response

(The authors gave the same response as above.)
